# Optimizing K_0.5_Na_0.5_NbO_3_ Single Crystal by Engineering Piezoelectric Anisotropy

**DOI:** 10.3390/nano11071753

**Published:** 2021-07-05

**Authors:** Weixiong Li, Chunxu Chen, Guangzhong Xie, Yuanjie Su

**Affiliations:** State Key Laboratory of Electronic Thin Films and Integrated Devices, School of Optoelectronic Science and Engineering, University of Electronic Science and Technology of China (UESTC), Chengdu 610054, China; weixiong_li@std.uestc.edu.cn (W.L.); cxchen@std.uestc.edu.cn (C.C.); gzxie@uestc.edu.cn (G.X.)

**Keywords:** piezoelectric, anisotropy, K_0.5_Na_0.5_NbO_3_, phase, temperature

## Abstract

K_0.5_Na_0.5_NbO_3_ is considered as one of the most promising lead-free piezoelectric ceramics in the field of wearable electronics because of its excellent piezoelectric properties and environmental friendliness. In this work, the temperature-dependent longitudinal piezoelectric coefficient $d_{33}^{*}$ was investigated in K_0.5_Na_0.5_NbO_3_ single crystals via the Landau–Ginzburg–Devonshire theory. Results show that the piezoelectric anisotropy varies with the temperature and the maximum of $d_{33max}^{*}$ deviates from the polar direction of the ferroelectric phase. In the tetragonal phase, $d_{33max}^{t^{*}}$ parallels with cubic polarization direction near the tetragonal-cubic transition region, and then gradually switches toward the nonpolar direction with decreasing temperatures. The maximum of $d_{33}^{o^{*}}$ in the orthorhombic phase reveals a distinct varying trend in different crystal planes. As for the rhombohedral phase, slight fluctuation of the maximum of $d_{33}^{r^{*}}$ was observed and delivered a more stable temperature-dependent maximum $d_{33max}^{r^{*}}$ and its corresponding angle *θ_max_* in comparison with tetragonal and orthorhombic phases. This work not only sheds some light on the temperature-dependent phase transitions, but also paves the way for the optimization of piezoelectric properties in piezoelectric materials and devices.

## 1. Introduction

With the gradual deepening and prosperity of the smart wearable industry revolution, piezoelectric-based flexible electronics have attracted considerable attention because of their promising applications in robotics [1], human–machine interaction (HMI) [2], energy harvesters [3], and internet of things (IOT) [4]. Lead-based perovskites, such as Pb(Zr_x_Ti_1−x_)O_3_ (PZT) [5] and Pb(Mg,Nb)O_3_ (PMN) [6] ceramics, possess huge piezoelectric properties but cause severe environmental and health concerns owing to their toxicity. As a promising alternative to lead zirconate titanate (PZT), the K_0.5_Na_0.5_NbO_3_, a lead-free ferroelectric material, has exhibited outstanding piezoelectric performance near the polymorphic phase boundary (PPB) [7] and attracted massive attention worldwide in recent years due to its environmental friendliness [8,9,10,11,12,13]. Owing to its unique merits of a high piezoelectric coefficient (d_33_), excellent ferroelectric properties, and a high Curie temperature (Tc = 420 °C), KNN has been widely utilized in energy-harvesting devices, transducers, actuators, and sensors [14,15,16,17,18,19,20,21]. Although pristine KNN ceramics possess relatively low piezoelectricity (d_33_~80 pC/N), they can be remarkably improved by tuning sintering conditions [22], domain engineering [23], phase boundary engineering [24,25], texturing [26], and so on. Furthermore, since the intrinsic piezoelectric response is intimately associated with spontaneous polarization rotation, the anisotropy of piezoelectric capability plays a crucial role in the application of piezoelectric materials [27]. Anisotropy of piezoelectric properties had attracted massive attention in the 1980s, and unprecedented large piezoelectric anisotropy was observed in lead titanate ceramics with random grain orientations [28,29]. However, limited works have focused on the piezoelectric anisotropy of K_0.5_Na_0.5_NbO_3_ single crystals.

The main purpose of this work is to study the piezoelectric anisotropy of K_0.5_Na_0.5_NbO_3_ single crystals as a function of temperature and to unravel the impact of phase transitions on the orientation and amplitude of the longitudinal piezoelectric coefficient. Landau–Ginzburg–Devonshire (LGD) theory was utilized to calculate the three-dimensional surface of the longitudinal piezoelectric coefficient $d_{33}^{*}$ for KNN single crystals in three ferroelectric phases as a function of temperature. Temperature-dependent free energy and spontaneous polarization of KNN has also been investigated to interpret the derivation of piezoelectric coefficients stemmed from temperature variation.

## 2. Materials and Methods

The ferroelectric capabilities of KNN single crystals were systematically investigated through the Landau–Ginzburg–Devonshire (LGD) function. To achieve more convenience in expressing piezoelectric coefficients in light of the coordinate system for each ferroelectric phase, the cubic paraelectric phase was selected as the reference. The thermodynamic potential function G of the KNN single crystal can be written as [30,31]:(1)$$G(\sigma ,E,T)=f_{LGD}+f_{elastic}+f_{electric}$$
where the Landau energy density is given by
(2)$$\begin{array}{cc} f_{LGD} & =\alpha_{1}(P_{1}^{2}+P_{2}^{2}+P_{3}^{2})+\alpha_{11}(P_{1}^{4}+P_{2}^{4}+P_{3}^{4})+\alpha_{12}(P_{1}^{2}P_{2}^{2}+P_{1}^{2}P_{3}^{2}+P_{2}^{2}P_{3}^{2})\\ & +\alpha_{111}(P_{1}^{6}+P_{2}^{6}+P_{3}^{6})+\alpha_{112}(P_{1}^{2}(P_{2}^{4}+P_{3}^{4})+P_{2}^{2}(P_{3}^{4}+P_{1}^{4})+P_{3}^{2}(P_{1}^{4}+P_{2}^{4}))\\ & +\alpha_{123}P_{1}^{2}P_{2}^{2}P_{3}^{2}+\alpha_{1111}(P_{1}^{8}+P_{2}^{8}+P_{3}^{8})+\alpha_{1122}(P_{1}^{4}P_{2}^{4}+P_{1}^{4}P_{3}^{4}+P_{2}^{4}P_{3}^{4})\\ & +\alpha_{1112}(P_{1}^{6}(P_{2}^{2}+P_{3}^{2})+P_{2}^{6}(P_{3}^{2}+P_{1}^{2})+P_{3}^{6}(P_{1}^{2}+P_{2}^{2}))\\ & +\alpha_{1123}(P_{1}^{4}P_{2}^{2}P_{3}^{2}+P_{2}^{4}P_{3}^{2}P_{1}^{2}+P_{3}^{4}P_{1}^{2}P_{2}^{2})\\ & -\frac{1}{2}s_{11}(\sigma_{1}^{2}+\sigma_{2}^{2}+\sigma_{3}^{2})-s_{12}(\sigma_{1}\sigma_{2}+\sigma_{1}\sigma_{3}+\sigma_{2}\sigma_{3})\\ & -\frac{1}{2}s_{44}(\sigma_{4}^{2}+\sigma_{5}^{2}+\sigma_{6}^{2})-Q_{11}(\sigma_{1}P_{1}^{2}+\sigma_{2}P_{2}^{2}+\sigma_{3}P_{3}^{2})\\ & -Q_{12}(\sigma_{1}(P_{2}^{2}+P_{3}^{2})+\sigma_{2}(P_{3}^{2}+P_{1}^{2})+\sigma_{3}(P_{1}^{2}+P_{2}^{2}))\\ & -Q_{44}(\sigma_{4}P_{2}^{2}P_{3}^{3}+\sigma_{5}P_{1}P_{3}+\sigma_{6}P_{1}P_{2}) \end{array}$$

Here, *α* denotes the Landau coefficients determined under the stress-free condition [28], *σ_i_* denotes the *i_th_* component of stress in Voigt notation, *s_11_*, *s_12_*, and *s_44_* denote the elastic compliance constants of a cubic phase [29,30,31], and *Q*_11_, *Q*_12_, and *Q*_44_ denote the corresponding electrostrictive coefficients between polarization and stress [32,33].

In this work, $P_{ijP}^{c}$, $\eta_{ijP}^{c}$, and $d_{ijP}^{c}$ denote the polarization, dielectric susceptibility, and piezoelectric coefficient in the KNN single crystal for each phase, respectively. For studying the orientational dependence of piezoelectric coefficients, Euler angle (φ,θ,ψ) is utilized to quantitatively describe the rotation in terms of ferroelectric phase coordinates.

The Landau coefficients, elastic compliance constants, and electrostrictive coefficients are taken from Jianjun’s previous work [34]. The dielectric constant possesses a positive proportional relationship with relative dielectric stiffness ($\varepsilon_{ij}^{p}=1+\eta_{ij}^{p}\approx \eta_{ij}^{p}$). K_0.5_Na_0.5_NbO_3_ endures a series of phase transition (cubic→tetragonal→orthorhombic→rhombohedral) in the process of cooling from the paraelectric phase. The following relations were utilized to calculate the temperature-dependence of piezoelectric coefficients $d_{ij}^{p}$ and dielectric susceptibility coefficients $\eta_{ij}^{p}$ of KNN crystals as a function of spontaneous polarization,
(3)$$\chi_{ij}=\varepsilon_{0}\partial^{2}f_{LGD}/\partial P_{i}\partial P_{j}(i,j=1,2,3)$$
(4)$$\eta_{ij}=A_{ji}/\Delta (i,j=1,2,3)$$
(5)$$g_{ij}=-\partial^{2}f_{LGD}/\partial P_{i}\partial \sigma_{j}$$
(6)$$d_{ij}=\varepsilon_{0}\eta_{ik}g_{kj}$$
where *A_ji_* and Δ refer to the cofactor and determinant of the *χ_ij_* matrix.

## 3. Results and Discussion

The LGD-free energy density $f_{LGD}$ of the tetragonal phase, orthorhombic phase, and rhombohedral phases, respectively, is plotted as a function of polarization at various temperatures, as shown in Figure 1a–c.

Figure 2 and Figure 3, respectively, elucidate the calculated dielectric susceptibility coefficients $\eta_{ij}^{p}$ and piezoelectric coefficients $d_{ij}^{p}$ for K_0.5_Na_0.5_NbO_3_ single crystals as a function of temperature in all three ferroelectric phases, respectively. 

According to the Landau–Ginsburg–Devonshire theory, the polarization, i.e., the second derivative of thermodynamic potential function G, can be acquired as a function of temperature by minimizing the total free energy in terms of polarization. Figure 4 illustrates the temperature-dependent spontaneous polarization of K_0.5_Na_0.5_NbO_3_ single crystals in the three phases. It can be clearly seen that polarization goes up with the cooling process for all the ferroelectric phases. Sudden rises were observed at the regions of phase transitions, where 648 K for cubic to tetragonal, 469 K for tetragonal to orthorhombic, and 130 K for orthorhombic to rhombohedral, which is consistent with the experimental results of 694 K, 468 K and 125 K, respectively, from Egerton et al. [35]. It is worth noting that the calculated polarization becomes zero when the temperature is approaching Curie temperature, implying that the system switches to paraelectric phase. Figure 5 displays the free energy (ΔG) of K_0.5_Na_0.5_NbO_3_ single crystals in the three phases as a function of temperatures. 

The piezoelectric properties are proportional to the flattening of the free energy profile. It can be clearly seen that the tetragonal-cubic phase transition causing an enhanced $d_{33}^{t}$ (Figure 3) can also be explained by the flattening of the free energy profile (Figure 1a). Consequently, the delayering of the free energy profile favors the enhancement of dielectric susceptibility. It is obvious that in the orthorhombic phase, the increasing temperature flattens the LGD-free energy well and makes it shallower with the heating-up process (Figure 1b), giving rise to the increase in dielectric susceptibility, and thus the increase in its piezoelectric response (Figure 1b). As for the temperature-dependent free energy for the rhombohedral phase in Figure 1c, the delayering of the free energy arising from the temperature rising also contributes to the enhancement of piezoelectric coefficients lying along a no-polar direction (Figure 3). 

In the tetragonal phase, the value of $d_{33}^{t}$ for K_0.5_Na_0.5_NbO_3_ crystals in the rotated coordinate along an arbitrary direction can be expressed as:(7)$$d_{33}^{t*}(\theta )=\cos\theta (d_{15}^{t}\sin^{2}\theta +d_{31}^{t}\sin^{2}\theta +d_{33}^{t}\cos^{2}\theta )$$
where angle *θ* denotes rotation always from [100]^t^. In the tetragonal phase, (*P*_1_ = *P*_2_ = 0, $P_{3\;}{=P}_{3T}^{c}\;\neq \;0$).

Therefore, by using Equation (7), the three-dimensional profile of calculated $d_{33}^{t^{*}}\left(\theta \right)$ of the tetragonal phase for three selected temperatures 500, 550, and 600 K is displayed in Figure 6a–c, respectively. As the temperature goes down from the cubic phase to the orthorhomibic phase, the surface of $d_{33}^{t^{*}}\left(\theta \right)$ varies during the cooling process. The direction of the largest $d_{33max}^{t^{*}}\left(\theta \right)$ lies along [001]_c_ direction at 600 K then switches to *θ*_max_ = 32.5° at 550 K, and finally to *θ*_max_ = 48.6° at 500 K. 

Attributed to the expression of $d_{33}^{t^{*}}\left(\theta \right)$ in Equation (7), it is obvious that $d_{33}^{t^{*}}\left(\theta \right)$ is determined by three parameters $d_{33}^{t}$, $d_{31}^{t}$, and $d_{15}^{t}$.

As shown in Figure 3, $d_{33}^{t}$ and $d_{15}^{t}$ changes rapidly near the temperature range of tetragonal-cubic and tetragonal-orthorhombic phase transitions, respectively, while little change in $d_{31}^{t}$ was observed in comparison with $d_{33}^{t}$ and $d_{15}^{t}$. The $d_{15}^{t}$ behaves like the dielectric permittivity ($\eta_{11}^{t}$,$\eta_{22}^{t}$) in the cubic phase as a function of temperature and increases with the process of cooling down toward the ferroelectric phase. When the temperature rises toward to the cubic phase (Figure 3), $d_{33}^{t}$ surpasses $d_{15}^{t}$ obviously.

As shown in Figure 2, calculated dielectric constants $\eta_{11}^{t}$ and $\eta_{33}^{t}$ vary in opposite tendencies in the whole tetragonal phase temperature range, which gives rise to a maximum $d_{33max}^{t^{*}}\left(\theta \right)$ along the polar direction in the high-temperature range. As displayed in Figure 7 and Figure 8, the corresponding angle θ for the maximum value of $d_{33}^{t^{*}}$ varies as a function of temperatures, which clearly demonstrates the influence of temperature on the phase transition. It can be seen that the maximum $d_{33max}^{t^{*}}\left(\theta \right)$ stays along [001]^t^ at high temperatures but deviates from [001]^t^ to the nonpolar direction when the temperature goes down, near the orthorhombic-tetragonal point. At the temperature T = 450 K, the maximum $d_{33max}^{t^{*}}\left(\theta \right)$ = 521.9 pC/N lies along the direction defined by *θ*_max_ = 52.98°. The maximum $d_{33max}^{t^{*}}\left(\theta \right)$ decreases at first, and then rises with increasing temperature, leading to a minimum at 550 K. This is because $d_{15}^{t}$ keeps increasing while $d_{33}^{t^{*}}\left(\theta \right)$ keeps decreasing during the cooling process, leading to the orientation variation and amplitude change of maximum $d_{33max}^{t^{*}}\left(\theta \right)$. It is reported that the pure KNN at 433 K exhibits a piezoelectric coefficient of 108 pC/N [36], which is similar to our calculated results (Figure 7) along the polar direction at 500 K. As shown in Figure 8, the angle *θ*_max_ for the maximum $d_{33max}^{t^{*}}\left(\theta \right)$ in tetragonal K_0.5_Na_0.5_NbO_3_ deviates away from 0° once the temperature is below 560 K. It should be noted that the $d_{33max}^{t^{*}}\left(\theta \right)$ of KNbO_3_ (KNO) single crystals follows a similar tendency as that for the KNN single crystal because they have the same phase transition sequences and structures [37]. As for the PbTiO_3_ single crystal [38], the maximum $d_{33max}^{t^{*}}\left(\theta \right)$ lies along the polar direction at all temperatures because its shear coefficient is too small at all temperatures to rotate d_33_ away from the polar direction, which is quite different from the change trend for KNN.

In the orthorhombic phase, more complex behaviors were observed in the variation of piezoelectric coefficients as a function of temperature in K_0.5_Na_0.5_NbO_3_ (Figure 3). Compared with $d_{31}^{o}$*,*$\;d_{32}^{o}$, and $d_{33}^{o}$ that are relatively insensitive to temperature, the two different shear coefficients play a key role in piezoelectricity, where the shear piezoelectric coefficient $d_{15}^{o}$ declines with increasing temperature, while $d_{24}^{o}$ follows an opposite tendency, as shown in Figure 3. As a consequence, the piezoelectric coefficient $d_{33}^{o^{*}}$ in K_0.5_Na_0.5_NbO_3_ exhibits a more sophisticated temperature-dependent trend compared with the tetragonal phase:(8)$$d_{33}^{o*}(\phi ,\theta )=\cos\theta [(d_{15}^{o}+d_{31}^{o})\sin^{2}\theta \sin^{2}\phi +(d_{24}^{o}+d_{32}^{o})\sin^{2}\theta \cos^{2}\phi +d_{33}^{o}\cos^{2}\theta )]$$

For the orthorhombic phase with *P*_1_ = 0, $P_{2\;}{=P}_{3}{=P}_{3O}^{c}\;\neq \;0$, the surface of the piezoelectric coefficient at four chosen temperatures of 200 K, 250 K, 300 K, and 350 K are respectively shown in Figure 9a–d. With a decreasing temperature, the direction of the maximum $d_{33}^{o^{*}}$ was slightly changed from the polar direction [001]°. The distinct changing tendencies of $d_{15}^{o}$ and $d_{24}^{o}$ are apparently responsible for the 90° rotation of the direction of the maximum $d_{33}^{o^{*}}$ in the cooling process. For instance, at 200 K, the maximum $d_{33}^{o^{*}}$ of 157.204 pC/N lies along the direction defined by *θ* = 52.4° and *φ* = π/2 (Figure 9a), but switches to the direction defined by *θ* = 52.9° and *φ* = 0 at 350 K with an amplitude of 291 pC/N (Figure 9d). Note that the $d_{33}^{o^{*}}$ reaches up to 223.87 pC/N at 300 K, which is very closed to the experimental results of 218 pC/N, indicating the accuracy and reliability of our modeling and calculation [39].

As unraveled in Figure 10, Figure 11, Figure 12 and Figure 13, the temperature-dependent maximum $d_{33max}^{o^{*}}$ and its corresponding angle *θ*_max_ are illustrated in planes of *φ* = 0 and *φ* = π/2 to systematically indicate the piezoelectric anisotropy of the orthorhombic K_0.5_Na_0.5_NbO_3_, where the direction of the maximum $d_{33max}^{o^{*}}$ is rotated by 90° with an increasing temperature, and its amplitude attains the largest value in the high-temperature range when approaching the tetragonal phase.

According to the calculated LGD-free energy-polarization relationship as plotted in Figure 1b, the flattening of the LGD-free energy well arising from temperature growth enhances dielectric susceptibility, as well as piezoelectric response. This tendency further endows stronger effects on the piezoelectric response of KNN single crystals in the tetragonal-orthorhombic transition instead of the orthorhombic-rhombohedral transition.

As for the low-temperature rhombohedral phase, the orientation dependence of $d_{33}^{r^{*}}$ can be given by
(9)$$d_{33}^{r*}(\theta ,\psi )=d_{15}^{r}\cos\theta \sin^{2}\theta -d_{22}^{r}\sin^{3}\theta +d_{31}^{r}\sin^{2}\theta \cos\theta +d_{33}^{r}\cos^{3}\theta$$

As displayed in Figure 14a–c, similar three-dimensional surfaces of $d_{33}^{r^{*}}$ were observed at temperatures of 50 K, 75 K and 100 K, implying that the temperature variation in the rhombohedral phase causes less impact on the three-dimensional surface of $d_{33}^{r^{*}}$ in comparison with those for the tetragonal and orthorhombic phase. Since no lower-symmetry phase exists as the temperature is approaching toward 0 K, the rhombohedral phase is the most stable one, and the three-dimensional surface of $d_{33}^{r^{*}}$ remains unchanged when the K_0.5_Na_0.5_NbO_3_ single crystal was gradually frozen during the cooling process.

It is found that $d_{15}^{r}$, $d_{22}^{r}$ (negative value) are proportional to the temperature, while $d_{32}^{r}$ and $d_{33}^{r}$ are almost temperature independent in the rhombohedral phase (Figure 3). As a result, the three-dimensional surface of $d_{33}^{r}$ slightly varies with varying temperatures, as shown in Figure 14a–c. At 50 K, the maximum $d_{33}^{r^{*}}$ of 284.444 pC/N lies along the direction defined by *θ* = 60.9° and *φ* = 0 (Figure 14a), while at 100 K, the maximum $d_{33}^{r^{*}}$ switches to the direction defined by *θ* = 61.1° and *φ* = 0 with an amplitude of 327.5 pC/N (Figure 14c). The dependence of the piezoelectric coefficient $d_{33}^{r^{*}}$ in KNN on the *θ* under various temperatures is revealed in Figure 15. The direction of the maximum value of $d_{33}^{r^{*}}$ varies with rising temperatures from 0 K to 130 K in the rhombohedral phase (Figure 16). Both maximum $d_{33max}^{r^{*}}$ and its corresponding angle *θ*_max_ are proportional to the temperature. The angles *θ* are higher than *θ* = 54.73°, which is consistent with the [001]^c^ (or [111]_r_) direction. 

According to the expression of $d_{33}^{r^{*}}$ in Equation (9), $d_{33}^{r^{*}}$ reveals an asymmetry attribute with respect to the axis defined by *θ* = 90°. It is obvious that the enhanced dielectric susceptibility, which is perpendicular to the polar direction, renders the large change in $d_{15}^{r}$. The polarization rotation near the phase transition region gives rise to the increase of piezoelectric response upon the heating process, which brings about the rotation of the maximum $d_{33max}^{r^{*}}$ and the slight variation in $d_{33}^{r}$. Furthermore, with respect to the LGD-free energy profile (Figure 1c), the delayering of the free energy well with increasing temperatures indicates that the piezoelectric coefficients enhance toward a no-polar direction.

## 4. Conclusions

In summary, LGD thermodynamic theory was utilized to investigate the temperature-induced phase transition and evolution of three-dimensional $d_{33}^{*}$ surface in KNN single crystals. The dielectric softening along the direction perpendicular to the polarization axis is responsible for the direction change of the maximum $d_{33}^{*}$ under various temperatures. During the ferroelectric phase transition, the increase of shear piezoelectric coefficients renders a significant enhancement in $d_{33}^{*}$ along the non-polar direction. This work not only looks into the fundamental understanding of the temperature-dependent phase transitions, but also paves the way for the optimization of piezoelectric properties in ferroelectric materials.

## Figures and Tables

**Figure 1 nanomaterials-11-01753-f001:**
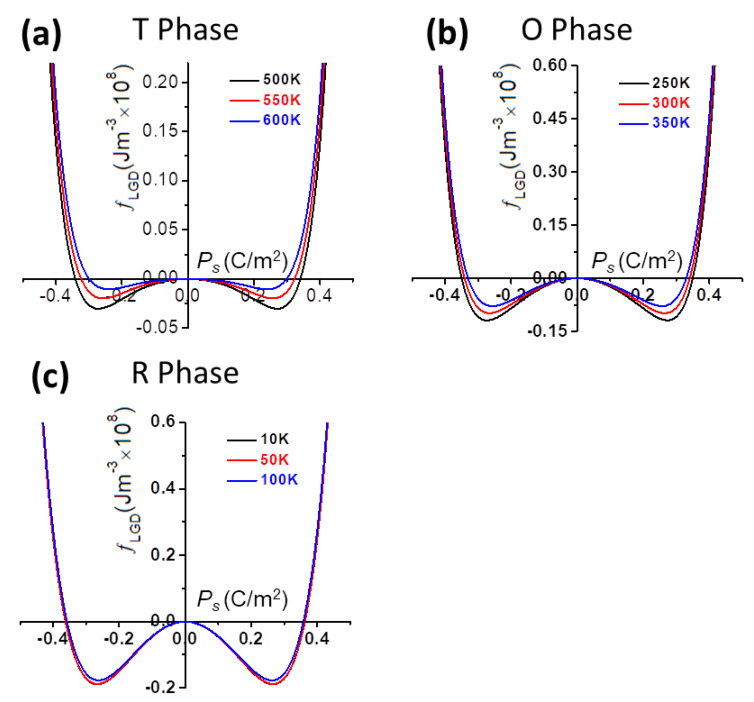
The Calculated LGD-free energy as a function of polarization (Ps) in (**a**) tetragonal phase, (**b**) orthorhombic phase, (**c**) rhombohedral phase at various temperatures.

**Figure 2 nanomaterials-11-01753-f002:**
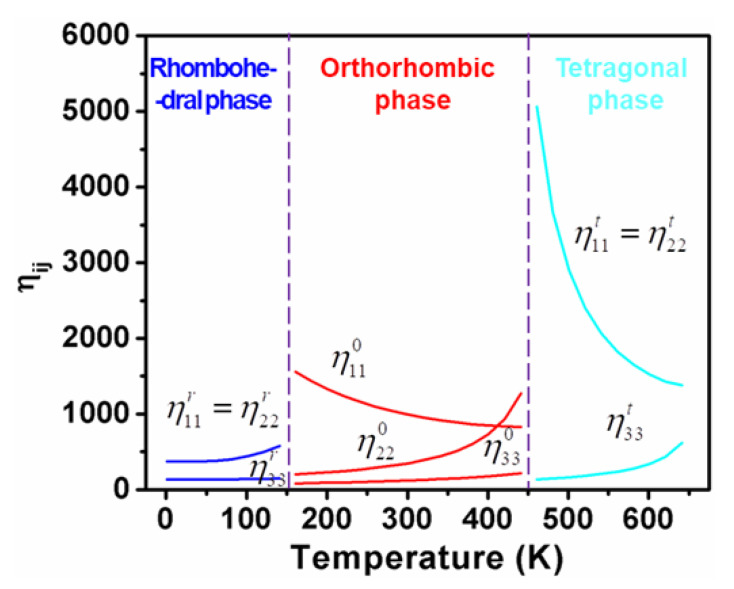
Calculated dielectric susceptibility coefficients for K_0.5_Na_0.5_NbO_3_ single crystals as functions of temperature in all three ferroelectric phases.

**Figure 3 nanomaterials-11-01753-f003:**
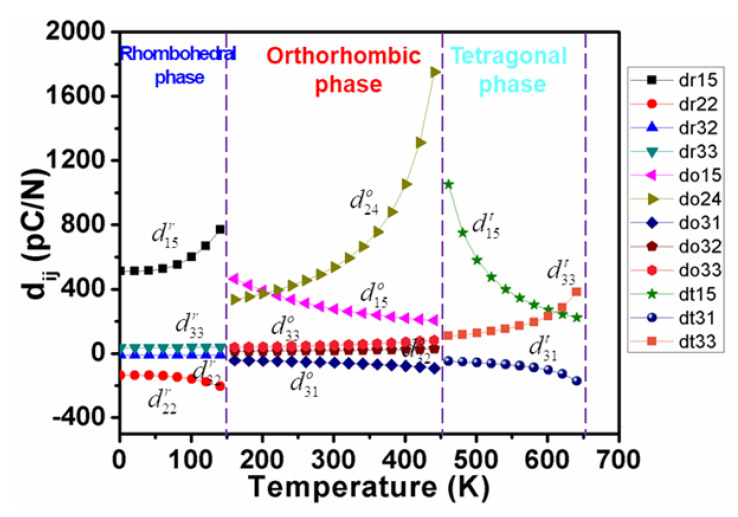
Calculated piezoelectric coefficients for K_0.5_Na_0.5_NbO_3_ single crystals as functions of temperature in all three ferroelectric phases.

**Figure 4 nanomaterials-11-01753-f004:**
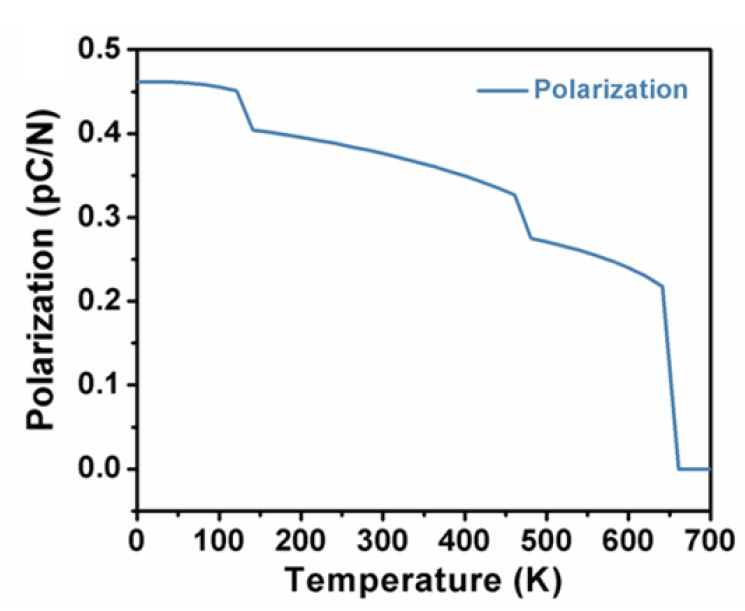
Spontaneous polarization of K_0.5_Na_0.5_NbO_3_ single crystals as a function of temperatures.

**Figure 5 nanomaterials-11-01753-f005:**
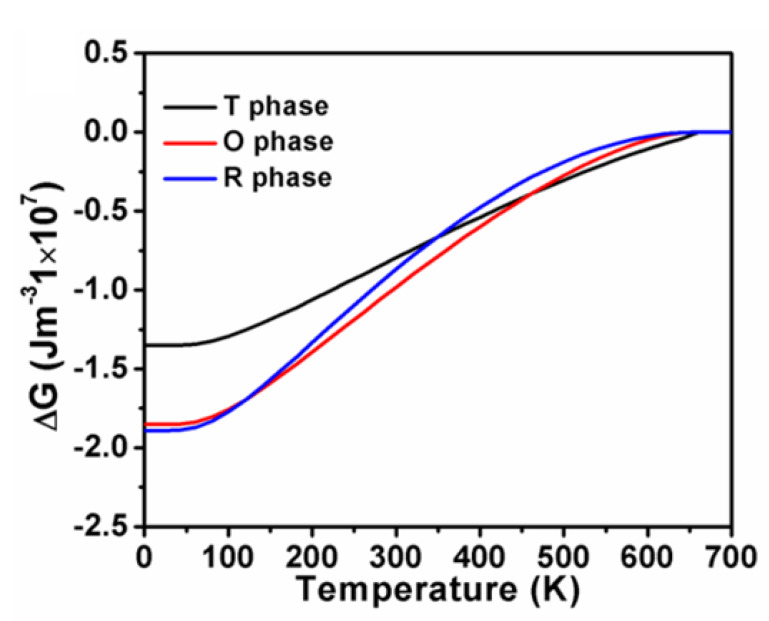
Free energy (ΔG) of K_0.5_Na_0.5_NbO_3_ single crystals in the three phases as a function of temperatures.

**Figure 6 nanomaterials-11-01753-f006:**
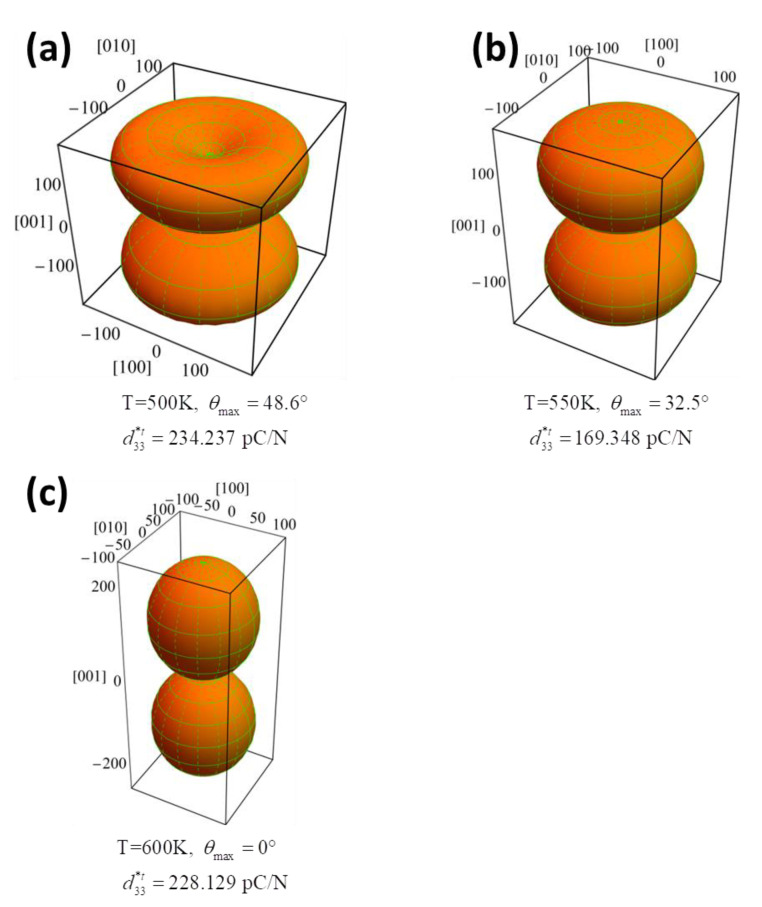
The orientation dependence of piezoelectric coefficient of K_0.5_Na_0.5_NbO_3_ single crystals in the tetragonal phases at the temperature of (**a**) 500 K, (**b**) 550 K, (**c**) 600 K.

**Figure 7 nanomaterials-11-01753-f007:**
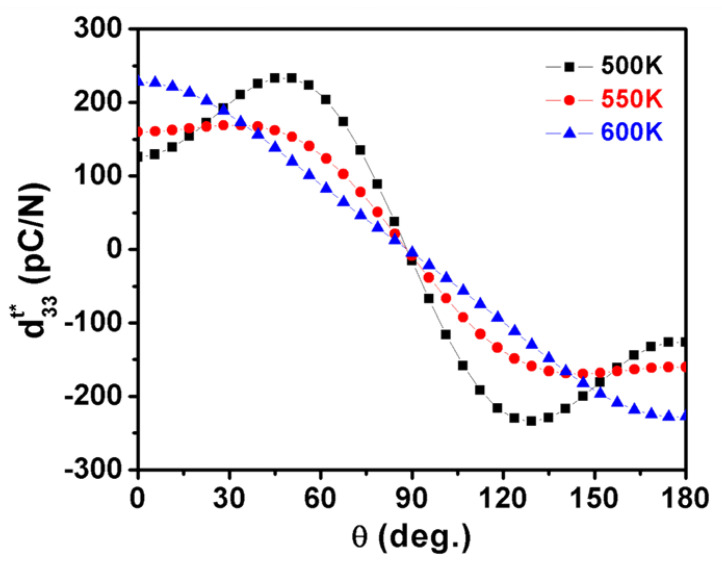
The piezoelectric coefficient in the tetragonal K_0.5_Na_0.5_NbO_3_ as a function of angle *θ* at various temperatures.

**Figure 8 nanomaterials-11-01753-f008:**
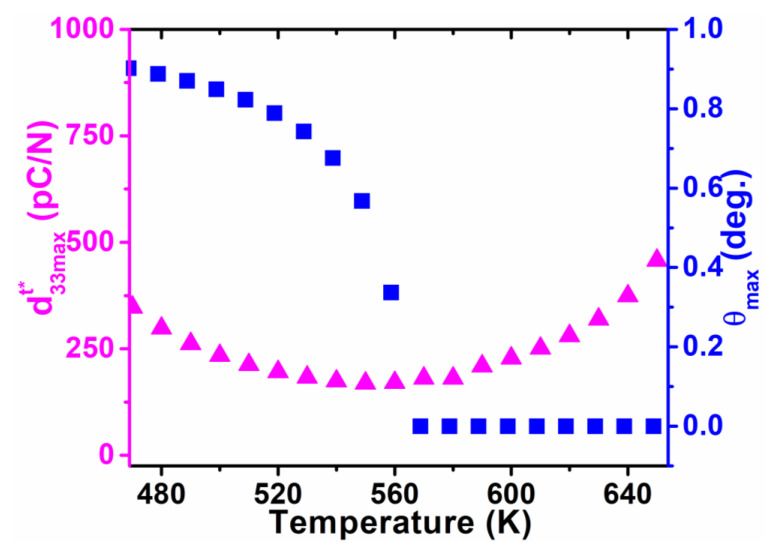
Maximum of piezoelectric coefficient and its corresponding angle *θ_max_* as a function of temperature for the tetragonal K_0.5_Na_0.5_NbO_3_.

**Figure 9 nanomaterials-11-01753-f009:**
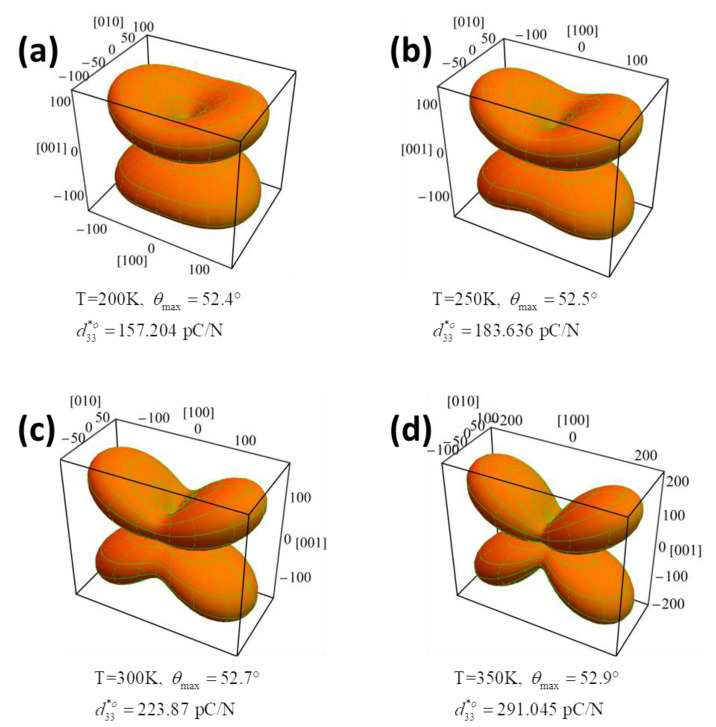
The orientation dependence of piezoelectric coefficient of K_0.5_Na_0.5_NbO_3_ single crystals in the orthorhombic phases at the temperature of (**a**) 200 K, (**b**) 250 K, (**c**) 300 K, (**d**) 350 K.

**Figure 10 nanomaterials-11-01753-f010:**
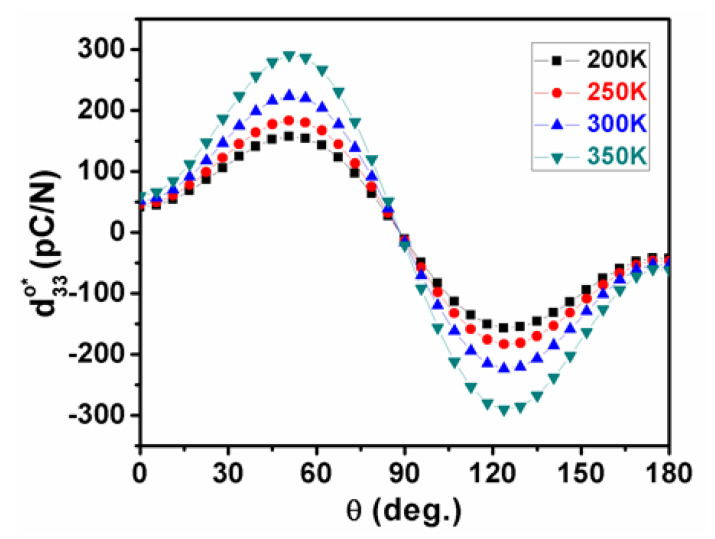
The piezoelectric coefficient in the orthorhombic K_0.5_Na_0.5_NbO_3_ as a function of angle *θ* under various temperatures in planes of *φ* = 0.

**Figure 11 nanomaterials-11-01753-f011:**
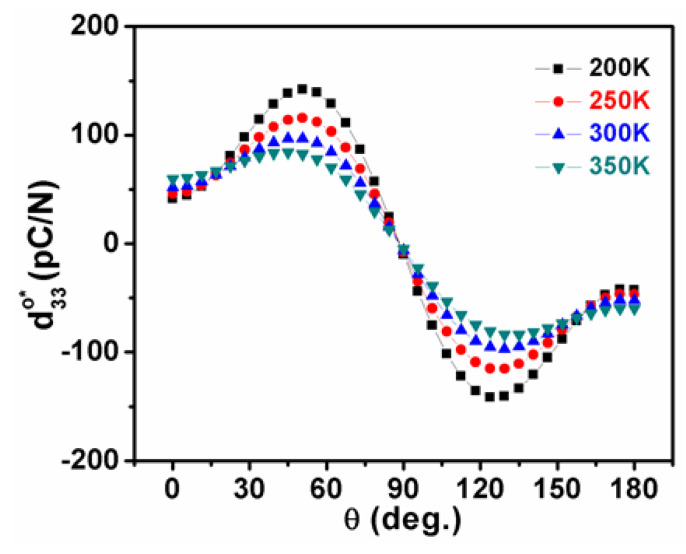
The piezoelectric coefficient in the orthorhombic K_0.5_Na_0.5_NbO_3_ as a function of angle *θ* under various temperatures in planes of *φ* = π/2.

**Figure 12 nanomaterials-11-01753-f012:**
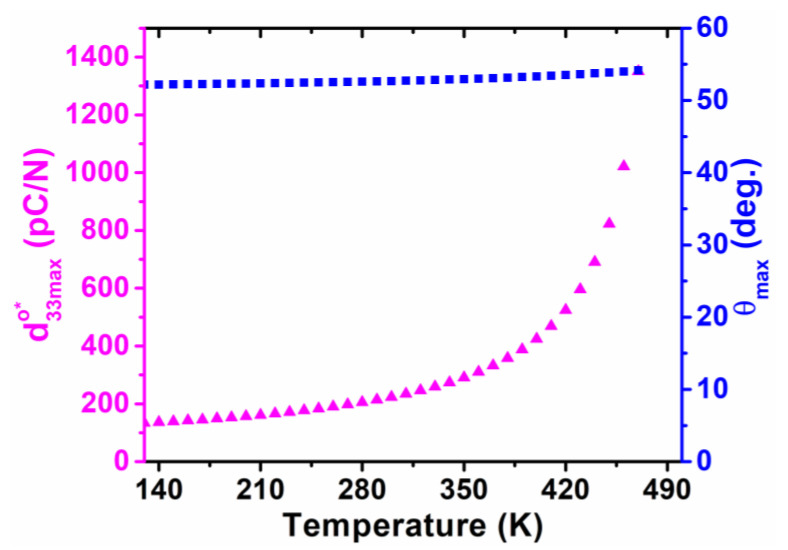
Maximum of piezoelectric coefficient and its corresponding angle *θ_max_* as a function of temperature for the orthorhombic K_0.5_Na_0.5_NbO_3_ in planes of *φ* = 0.

**Figure 13 nanomaterials-11-01753-f013:**
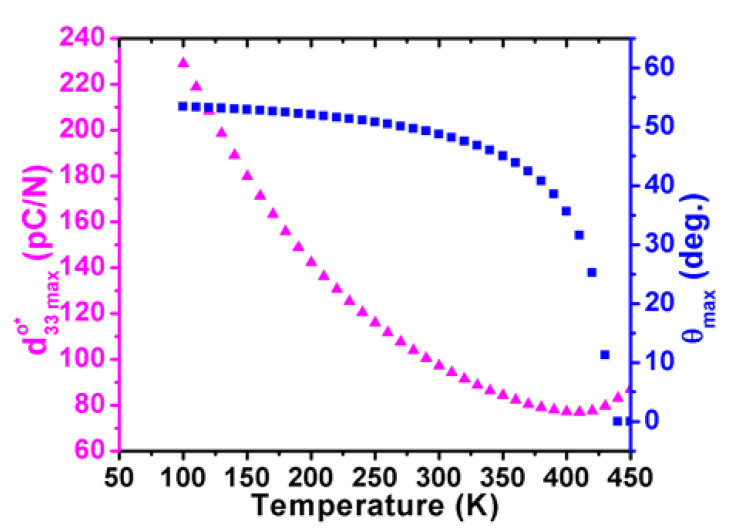
Maximum of piezoelectric coefficient and its corresponding angle *θ_max_* as a function of temperature for the orthorhombic K_0.5_Na_0.5_NbO_3_ in planes of *φ* = π/2.

**Figure 14 nanomaterials-11-01753-f014:**
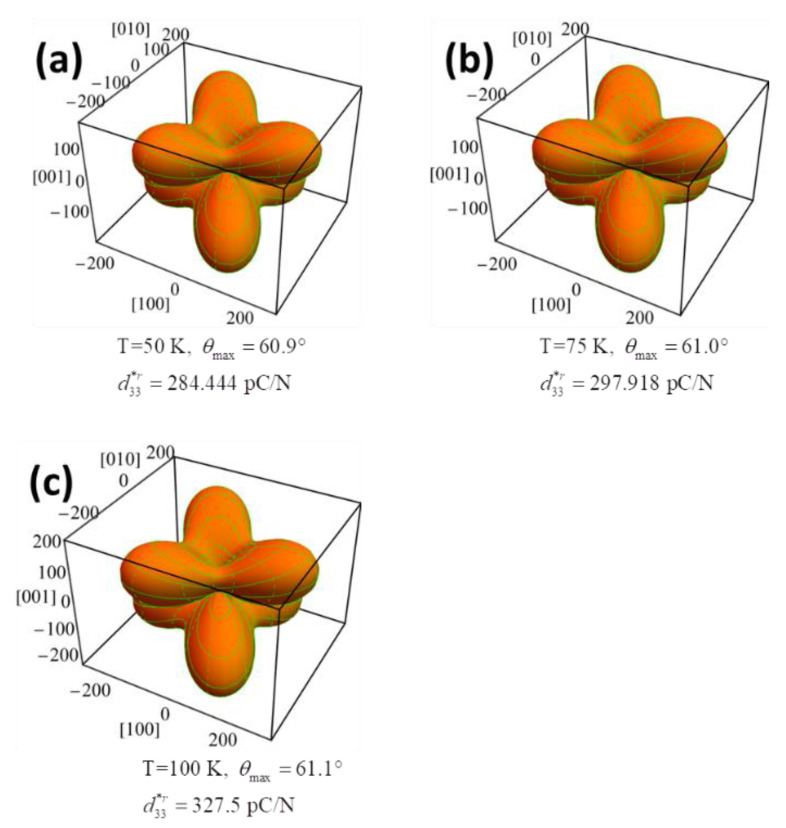
The orientation dependence of coefficient of K_0.5_Na_0.5_NbO_3_ single crystals in the rhombohedral phases at the temperature of (**a**) 50 K, (**b**) 75 K, (**c**) 100 K.

**Figure 15 nanomaterials-11-01753-f015:**
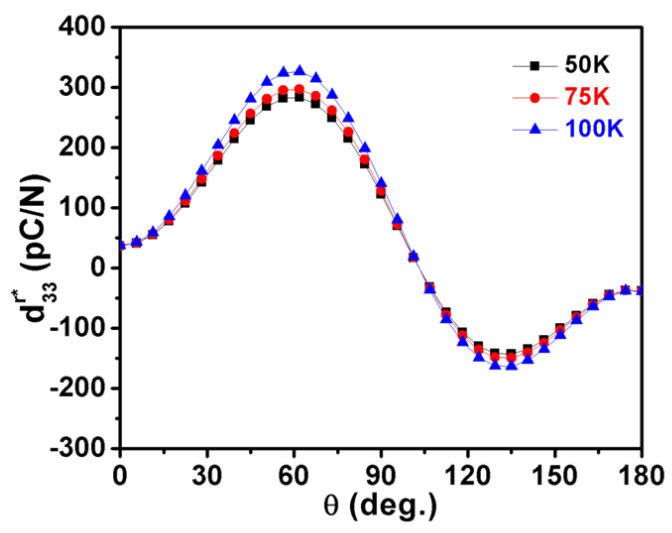
The piezoelectric coefficient in the rhombohedral K_0.5_Na_0.5_NbO_3_ as a function of angle *θ* at various temperatures.

**Figure 16 nanomaterials-11-01753-f016:**
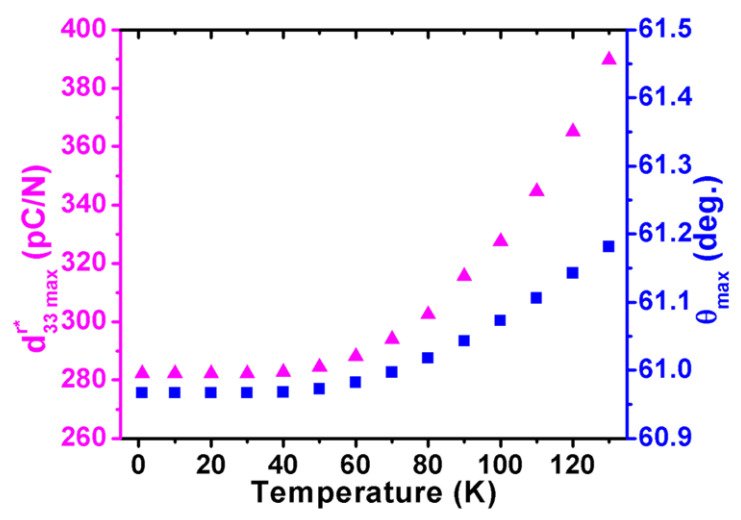
Maximum and its corresponding angle *θ_max_* as a function of temperature for the rhombohedral K_0.5_Na_0.5_NbO_3_.

## Data Availability

The data presented in this study are available on request from the corresponding author.

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
