# Peer review of "Optimizing K0.5Na0.5NbO3 Single Crystal by Engineering Piezoelectric Anisotropy"

_nanomaterials, 2021, doi:10.3390/nano11071753_

Round 1
Reviewer 1 Report
The manuscript by Li et al. (ID: nanomaterials-1261879) reports a detailed analysis of the piezoelectric anisotropy of K0.5Na0.5NbO3 (KNN) using the phenomenological Landau-Ginzburg-Devonshire (LGD) theory. KNN is an important lead-free piezoelectric material, which has attracted much attention over the past 15 years. The authors build up upon previous LGD works on this material, but at the same time present some new insights into the topic. Of particular interest is their detail analysis of the individual contributions to the overall piezoelectric response in the three phases. The work is thus of general interest to the community; however, before further consideration, the authors should address the criticism listed below. In addition, the presented work in my opinion does not justify the need for a “Communication article”, but seems more suitable for a “Regular article”.
- The title is inappropriate, as it implies that the authors have optimized an actual nanogenerator. This is a purely theoretical work and no work on nanogenerators was conducted. This word should thus be removed from the title. In addition, the title should include some wording that reveals that this is a LGD analysis and a theoretical work.
- In the Introduction (page 1) the authors relate the good properties of KNN to the existence of an MPB (line 35). While this was indeed assumed in early works on this system, it was later demonstrated that all the phase transitions in this system are of the polymorphic phase boundary type, not MPB (see for example here: https://doi.org/10.1557/mrs.2018.178).
- Since this work focuses on KNN single crystals, the authors should include a short description of these materials in the Introduction; here, relevant literature is missing. For example, they should report some of the seminal early works on these crystals (https://doi.org/10.1103/PhysRev.96.581), highest properties (https://doi.org/10.1039/C9TC05143K or https://doi.org/10.1016/j.actamat.2018.02.026), examples of applications (https://doi.org/10.1038/srep39679 or https://doi.org/10.1063/1.4990072), and recent review articles (https://doi.org/10.1557/jmr.2019.391). Also, an appropriate reference is missing when referring to the phase transitions of KNN (line 79, page 2).
- In Methods (line 60 of page 2), the authors cite the references from which the LGD coefficients were obtained (Refs. 22-26). However, they should be more specific and say which of these references was used or which coefficients were obtained from which paper.
- Please label the x axis in Figure 1 and also give units for axes in Figures 6, 9 and 14.
- Since this is a purely theoretical work, it would be important that the authors also briefly discuss the limitations of this theory. What assumptions were made and how do these need to be considered when comparing the results to real crystals? Also, it would be interesting to compare the obtained parameters to experimental results and comment on similarities/differences (the authors only compare the transition temperatures, but not other parameters).
- The captions of several figures are unclear or incorrect and should be revised. Figure 5 does not show “Spontaneous polarization”, in Figure 6 it is not specified which “coefficient” is shown, in Figure 8 the authors should state the “Maximum” of what parameter is shown.
- The comparison of KNN results to BTO and PTO on page 8 is interesting; however, it would be also interesting to compare the results to KNO, which is more similar to KNN than the other two materials. See for example: https://doi.org/10.1063/1.3511336
- The manuscript would be much improved, if the authors would compare their values of P and d33 to previous experimental reports. Also, it would be very helpful to the reader if the authors could give some summarizing guidelines for experimentalist at the end of the paper, stating which phases, conditions, and angles are most suitable for obtaining highest piezoelectric performance.
- The manuscript also contains many inconsistencies and irrational statements. For example, in the Abstract (lines 19-22) the authors seem to compare the rhombohedral phase to the rhombohedral phase (?). Line 143 on page 6 refers to °C, but I think they mean K. On page 5 they state that the O-T transition is at 469 K, but Figure 8 shows d33t values to lower temperatures; similarly, the R-O transition is reported to be at 130 K (page 5), but Figure 12 shows d33o values down to 100 K. Also, the authors should always refer to the first author, when referring to previous literature (e.g., line 77 of page 2). In addition, the authors should carefully revise the English language, as the manuscript contains many grammatical mistakes (in particular, the incorrect use of singular/plural).
Reviewer 2 Report
In this manuscript, the authors use the Landau-Ginsburg-Devonshire theory to investigate the piezoelectric anisotropy of a K0.5Na0.5NbO3 single crystal. They calculate the orientation dependence of the piezoelectric coefficients with temperature, and thus in the different crystallographic phases.
1- My main concern is that this work is very similar to the article of Liang et al. investigating the piezoelectric anisotropy of a KNbO3 single crystal [Liang, L., Li, Y. L., Hu, S. Y., Chen, L.-Q. & Lu, G.-H. Piezoelectric anisotropy of a KNbO3 single crystal. J. Appl. Phys. 108, 094111 (2010)]. There are huge similarities between the figures of both articles and the text (e.g. the abstract). Even if the composition investigated is not the same, i.e. K0.5Na0.5NbO3 instead of KNbO3, this has only little influence on the properties investigated because both systems are known to be extremely similar: for instance, Fig. 10 gives similar results to Fig. 8 of Liang. et al, Fig. 15 gives similar results to Fig. 10 of Liang. et al.
2- Furthermore, the framework of the calculation used in this manuscript is already well established for K0.5Na0.5NbO3:
- The free energy, dielectric susceptibility, piezoelectric coefficients, and polarization corresponding to Fig. 1-5 have already been published in a much more extensive way (since different compositions are reported) in Pohlmann, H., Wang, J.-J., Wang, B. & Chen, L.-Q. A thermodynamic potential and the temperature-composition phase diagram for single-crystalline K1-xNaxNbO3 (0 ≤ x ≤ 0.5). Appl. Phys. Lett. 110, 102906 (2017).
3- An extensive comparison with the piezoelectric coefficients experimentally measured in the literature would be welcome.
4- There are several small mistakes in the text:
- In Figs. 10-13, the caption is incorrect and should mention the “orthorhombic” phase instead of the “tetragonal” phase.
- In Figs. 15 and 16, the caption is incorrect and should mention the “rhombohedral” phase instead of the “tetragonal” phase.
- The caption of Fig. 4 corresponds to Fig. 5 and vice versa.
5- The authors mention “nanogenerator” in the title of their manuscript, but never in the main text.
Round 2
Reviewer 1 Report
The authors have prepared a point-to-point response to Reviewers´ comments. Although most criticism was adequately addressed, the changes were not always appropriately implemented in the revised manuscript or the changes made were sloppy. Below is a list of my remarks, which should still be carefully addressed.
- The authors agreed to correct the title, but the revised manuscript still had the old title. Please correct.
- Reply to Comment #2 (page 1 of revised manuscript): The “polymorphic phase boundary” should be abbreviated as PPB, not MPB.
- The references in the text are not ordered. For example, Refs. 14-19 do not appear on Page 1. Please carefully revise the numbers.
- Grammatical errors exist (in particular in the newly-added text) and should be corrected before publication.
- The comparison with KNO should be improved (page 8). Firstly, this material should be written with full formula, not abbreviation. Secondly, it is hard to believe that KNO behaves different from KNN, since they have same phase transitions sequence and structures. At this point, the authors should do a careful comparison to previous works on KNO (see the comments from Reviewer #2) and KNN. Just citing previous work, as the authors stated in their response to Reviewer #2 is not sufficient.
- Reply to my previous comment #8: It is not clear why the authors mentioned thin films in the revised manuscript. This work is clearly focused on the general aspect of KNN material, not on their thin film form, since the influence of strain/stress on the parameters was not studied (which would be crucial for thin films). I thus recommend to the authors to keep the discussion general for all material forms.
- Parts of the abstract are still incomprehensible. For example, the sentence in lines 18-21 states that some properties of the rhombohedral phase are more stable in comparison to the rhombohedral phase.
Reviewer 2 Report
1- While the authors have made a few changes to their manuscript, I still believe that an extensive comparison with the piezoelectric coefficients experimentally measured in the literature is missing. Even if the motivation of the authors seems to be on thin films, the theoretical values obtained could be compared to experimental values reported for single crystals and textured ceramics.
2- On line 35, the authors now mention the “polymorphic phase boundary (MPB)”, it should be the “polymorphic phase boundary (PPB)”.
3- The authors now mention that “In comparison with BaTiO3, PbTiO3 and KNO [36], the angle θmax for the maximum d33maxt*(θ) in tetragonal K0.5Na0.5NbO3 deviates away from 0° once the temperature below 560 K (Fig. 8).” However, ref. 36 is only about KNO. The authors should thus cite references for BaTiO3 and PbTiO3 as well. Furthermore, for KNO, the angle θmax for the maximum d33maxt*(θ) in the tetragonal phase also deviates away from 0° once the temperature is below 560 K [Liang, L., Li, Y. L., Hu, S. Y., Chen, L.-Q. & Lu, G.-H. Piezoelectric anisotropy of a KNbO3 single crystal. J. Appl. Phys. 108, 094111 (2010)]. As such it is misleading to cite it along with BaTiO3 and PbTiO3.
